# Circadian Misalignment Induced by Chronic Night Shift Work Promotes Endoplasmic Reticulum Stress Activation Impacting Directly on Human Metabolism

**DOI:** 10.3390/biology10030197

**Published:** 2021-03-05

**Authors:** Rafael Ferraz-Bannitz, Rebeca A. Beraldo, Priscila Oliveira Coelho, Ayrton C. Moreira, Margaret Castro, Maria Cristina Foss-Freitas

**Affiliations:** 1Division of Endocrinology and Metabolism, Department of Internal Medicine, Ribeirao Preto Medical School, Avenida Bandeirantes, 3900—Vila Monte Alegre, 14049-900 Ribeirao Preto/SP, Brazil; rebecaberaldo@yahoo.com.br (R.A.B.); acmoreir@fmrp.usp.br (A.C.M.); castrom@fmrp.usp.br (M.C.); crisfoss@fmrp.usp.br (M.C.F.-F.); 2Department of Biochemistry, Ribeirao Preto Medical School, University of Sao Paulo, Avenida Bandeirantes, 3900—Vila Monte Alegre, 14049-900 Ribeirao Preto/SP, Brazil; priscila_coelho6@hotmail.com

**Keywords:** endoplasmic reticulum stress, CLOCK genes, night shift, circadian rhythm, metabolic syndrome

## Abstract

**Simple Summary:**

The demands of modern society have made shift work a necessity. Night work is associated with an increased risk of metabolic problems such as obesity and diabetes, which is mainly due to the misalignment of circadian rhythms that play a crucial role in many biological processes. This study performed clinical, anthropometric, and molecular analyses on 40 hospital workers who work day or night. We demonstrated that night workers had increased glucose levels, triglycerides, waist circumference, and blood pressure compared to day workers. Surprisingly, we report that night workers have significant changes in the expression of circadian clock genes and an up-regulation of genes related to endoplasmic reticulum stress (ERS). These findings provide new insights into the effects of night shift work on the expression of circadian cycle genes and ERS activation, leading to metabolic stress and the development of metabolic diseases associated with night work.

**Abstract:**

Night work has become necessary in our modern society. However, sleep deprivation induces a circadian misalignment that effectively contributes to the development of diseases associated with metabolic syndrome, such as obesity and diabetes. Here, we evaluated the pattern of circadian clock genes and endoplasmic reticulum stress (ERS) genes in addition to metabolic and anthropometric measures in subjects that work during a nocturnal period compared with day workers. We study 20 night workers (NW) and 20 day workers (DW) submitted to a work schedule of 12 h of work for 36 h of rest for at least 5 years in a hospital. The present report shows that NW have increased fasting blood glucose, glycated hemoglobin (HbA1c), triglycerides, and low-density lipoprotein (LDL)-cholesterol levels, and lower high-density lipoprotein (HDL)-cholesterol levels compared to DW. In addition, we observed that waist circumference (WC), waist–hip ratio (WHR), and systemic blood pressure are also increased in NW. Interestingly, gene expression analysis showed changes in CLOCK gene expression in peripheral blood mononuclear cells (PBMC) samples of NW compared to the DW, evidencing a peripheral circadian misalignment. This metabolic adaptation was accompanied by the up-regulation of many genes of ERS in NW. These findings support the hypothesis that night shift work results in disturbed glycemic and lipid control and affects the circadian cycle through the deregulation of peripheral CLOCK genes, which is possibly due to the activation of ERS. Thus, night work induces important metabolic changes that increase the risk of developing metabolic syndrome.

## 1. Introduction

Sleep deprivation in humans is associated with the prevalence of metabolic syndrome [1,2,3,4] and increased risk of mortality [5]. Healthy individuals are those who present the temporal relations between their organism and the external environment in synchrony with their circadian cycles [6]. In this way, the expression of biological rhythms happens through the interaction of endogenous and exogenous factors. Biological rhythms can modulate many physiological processes in mammals; also, several genes are under the influence of circadian control in different tissues [7,8]. The act of working is intrinsic to humans; however, night work has become indispensable to the globalized economy, especially in health sectors such as hospitals. Many shift workers, especially night workers, have unhealthy eating habits and sedentary lifestyle [9], further aggravating metabolic changes. The reduction of the sleep period is directly associated with the increase of the body mass index (BMI) and food consumption in a way that is dependent on the alterations of the CLOCK genes expression [10,11]. In addition to changing habits, sleep deprivation affects daily rhythms, which are controlled by self-regulating biological oscillators. The molecular components present in the circadian cycle pathway are mainly *PER1* and *CRY1* being negative regulators and *CLOCK* and *BMAL1* as positive regulators [12,13,14,15].

It is known that endoplasmic reticulum stress (ERS) activation is a strong molecular link between obesity, lipodystrophy, functional insulin damage, and the development of type 2 diabetes mellitus [16,17]. The endoplasmic reticulum (E.R.) is an organelle formed by a membranous network located in the cellular cytoplasm and responsible for the synthesis and processing of secretory and membrane proteins. The pancreatic E.R. kinase (EIF2AK3) is a transmembrane protein kinase of the E.R. that phosphorylates Eukaryotic Initiation Factor 2α (eIF2 α) in response to the ERS signal. Therefore, the phosphorylation of EIF2AK3 and eIF2α is the main sign of the presence of ERS [18,19]. In this case, when the ERS is active by EIF2AK3, this signal is directed to activating Activator Transcription Factor 4 (ATF4), resulting in the reduction of protein synthesis and avoiding the accumulation of new malformed proteins. Furthermore, the 78-kD glucose-regulated/binding immunoglobulin protein (GRP78) is an E.R. chaperone whose gene expression is increased with ERS [20]. Nevertheless, it is not clear that metabolic changes associated with reduced sleep time result only from changes in macronutrients or energy intake.

Our hypothesis in this study is that night shift work induces important metabolic disorders such as increased levels of fasting glucose, Hb1Ac, triglycerides, and LDL, contributing to the development of the metabolic syndrome. Furthermore, we identify that night shift work affects peripheral circadian clock genes and induces the activation of ERS in night shift workers compared to day workers.

## 2. Materials and Methods

### 2.1. Participants and Study Design

Peripheral blood mononuclear cells (PBMC), serum, and plasma were obtained from 40 subjects, 20 day workers (10 men and 10 women) and 20 night workers (10 men and 10 women) in the intensive care unit (I.C.U.) of a large hospital in Ribeirao Preto, São Paulo, Brazil. We selected workers with no history of chronic diseases. All individuals had a Bachelor’s degree, including physicians, nurses, and biomedical staff. The minimum time for individuals working at night was 5 years, while the minimum time for individuals working in the daytime period was 6 years. The working time was 12 h, being day workers from 0700 h until 1900 h and night workers from 1900 h until 0700 h. Blood samples were collected in an 8-h fasting state at 0700 h and 1900 h on the same day of work. Blood samples from daytime workers were collected at the beginning (0700 h) and at the end (1900 h) of the workday. Night workers’ blood samples were collected at the beginning (1900 h) and end (0700 h) of the workday. The local Ethics Committee approved the study’s protocol (Trial Registration: C.A.A.E., 82573317.3.0000.5440. Registered 07 May 2018, http://plataformabrasil.saude.gov.br/visao/publico/indexPublico.jsf) and informed written consent was obtained from all patients.

Data on usual sleep duration during the week and working day in hours were obtained from the participants using a questionnaire. 

### 2.2. Biochemical Analysis

A blood sample was collected at 0700 h after an overnight fast of 8 h. Plasma glucose levels and Hb1Ac were measured using COBAS INTEGRA 400 plus (Roche^®^, Indianapolis, IN, USA). Cortisol levels were measured using a radioimmunoassay by Tri-carb 2100 tr Liquid Scintillation Analyzer (Packard^®^, Conroe, TX, USA). Serum samples for fasting lipids were analyzed, and for the present study, serum levels of total cholesterol, LDL, HDL, and triglycerides were measured as well using COBAS INTEGRA 400 plus (Roche^®^, Indianapolis, IN, USA). 

### 2.3. Anthropometric Indicators and Body Composition

Body weight (kg) was measured with an electronic Filizola scale of platform type with a maximum capacity of 300 kg and precision of 0.1 kg. Height was measured with a stadiometer with 0.1 cm precision. BMI was calculated as weight (kg) divided by height (m) squared.

The circumferences were performed using a metal measuring tape, Sanny, accurate to 0.1 cm and a maximum length of 2 m.

Waist circumference (WC) was performed midway between the inferior margin of the last rib and the crest of the ilium in a horizontal plane.

The body fat percentage was obtained using a bioelectrical impedance analyzer (B.I.A.) (Biodynamics^®^ 450 model).

### 2.4. R.N.A. Extract and Gene Expression Analysis

Cells were isolated using Ficoll-Hypaque (Sigma^®^, St. Louis, MO, USA). Total R.N.A. was extracted using Trizol reagent (Life Technologies^®^, Carlsbad, CA, USA), following the manufacturer’s instructions and confirmed to be free of proteins or phenol using U.V. spectrophotometry. The cDNA synthesis was conducted using the iScript cDNA Synthesis Kit (Bio-Rad^®^, Hercules, CA, USA) using 1 μg of total R.N.A. To assess gene expression of genes related to the circadian cycle (CLOCK genes) and endoplasmic reticulum stress (ERS), we used blood samples (PBMC) collected at 0700 h and 1900 h. The gene expression was normalized to GAPDH expression and data are presented as fold change over housekeeping gene.

### 2.5. RT-qPCR

Then, the gene expression rate was evaluated by quantitative real-time PCR (qPCR). Each reaction mixture containing 250 nM of each primer (sense and antisense), 25 ng of cDNA, and SsoFast EvaGreen Supermix (Bio-Rad^®^) in a final volume of 10 μL was analyzed in a CFX96 Touch™ Real-Time PCR Detection System (Bio-Rad^®^) under the following amplification conditions: 50 °C-2 min, 95 °C-10 min, 40 cycles of 95 °C-15 s, 60 °C-20 s, and 72 °C-30 s and data were analyzed with the 2^−ΔΔCT^ method. The primers’ sequence for the genes of interest and a housekeeping gene (GAPDH) as the endogenous control used in our experiments are shown in Appendix A. 

### 2.6. Statistical Analysis

Results are expressed as the mean ± standard deviation. We used the Student *t*-test to compare day shift vs. night shift or one-way repeated measures ANOVA as appropriate. We computed the CV (coefficient of variation, the standard deviation divided by the mean) to demonstrate the variability of cortisol levels between groups. We used the Graphpad Prism^®^ 6.5 (Mac) of statistical tools to analyze and test the metabolic and gene expression data. Statistical significance was considered when *p* < 0.05.

## 3. Results

Demographic information for the cohort is provided in Table 1. We selected 20 individuals working during the day (DW) (10F, 10 M; mean age 38 ± 6.8; body weight 70.1 ± 12.4; height 1.64 ± 0.07) and 20 individuals working at night (NW) (10F, 10 M; mean age 40 ± 4.9; body weight 77.3 ± 10.0; height 1.72 ± 0.08). The groups did not differ significantly by age (*p* = 0.1858), gender (*p* = 0.9999), body weight (*p* = 0.0511); however, the NW group was taller than the DW group (*p* = 0.0041) (Table 1). Sleep time analyses revealed that NW group participants slept more than the DW group on working days (9.1 h ± 0.6 vs. 8.1 h ± 0.5, (*p* = 0.0001), Table 1). However, on free days, participants in the NW group had lower sleep time as compared with DW (8.2 h ± 0.4 vs. 8.5 h ± 0.5, (*p* = 0.0210), Table 1).

### 3.1. Metabolic Parameters and Anthropometric Measures

Biochemical data are shown in Figure 1. Fasting blood glucose and glycated hemoglobin (HbA1c) was 14% (*p* = 0.0024) and 14.6% (*p* < 0.0001) higher, respectively, in NW in comparison to DW (Figure 1a,b). Evaluating the participant’s lipid profile shows no differences in the total cholesterol concentration (*p* = 0.2153) (Figure 1c). Nevertheless, the triglyceride levels of the NW group were 45.7% (*p* = 0.0359) higher compared to the DW group (Figure 1d). Plasma HDL cholesterol was significantly decreased, 17.3% (*p* = 0.0456) in the NW group (Figure 1e). LDL cholesterol levels were 25.8% (*p* < 0.0001) increased in the NW group compared to DW (Figure 1f). There was no difference in C Reactive Protein (C.R.P.) (*p* = 0.9099) between the groups (Figure 1g). The serum cortisol concentration was significantly higher in both groups at 0700 h (DW: 0700 h vs. 1900 h (*p* = 0.0395); NW: 0700 h vs. 1900 h (*p* = 0.0001)), comparing samples collected at 0700 h vs. 1900 h. However, we observed greater variability in cortisol among the NW group both times (coefficient of variation of NW 68.1% vs. DW 40.6%) and 1900 h (CV NW 90.2% vs. DW 60.6%) (Figure 1h). 

The blood pressure assessment showed that NW has an increase in systolic and diastolic pressure compared to DW (4.6% (*p* = 0.0496) and 9.7% (*p* = 0.0172), respectively) (Figure 2a). The body mass index (BMI) did not differ between groups (*p* = 0.7801) (Figure 2b). However, the waist circumference was significantly increased in the NW group by 9.8% (*p* = 0.0010) (Figure 2c). When analyzing the waist–hip ratio (WHR), we noticed an increase of 4.6% (*p* = 0.0153) in the NW group compared to the DW group (Figure 2d). Lastly, there was no difference in the total fat mass between NW and DW (*p* = 0.0887) (Figure 2e). No significant differences were found between genders.

Moreover, the correlation analyses between body weight and plasma metabolites concentration showed a significant negative correlation between body weight and triglycerides levels in the NW group (r = −0.51; *p* = 0.0201) (Appendix A).

### 3.2. CLOCK Genes and the Circadian Rhythm

The study of the mRNA expression of clock genes from PBMC samples collected at 0700 h and 1900 h indicated that in the DW group, *BMAL1* expression presented a robust reduction at 1900 h (*p* < 0.0001) but no change in *CLOCK* expression (*p* = 0.3980) (Figure 3a). However, the expression of *CRY1* and *PER1* showed an increase at 1900 h compared to 0700 h (*p* < 0.0001; *p* = 0.0330 respectively) (Figure 3a). In the NW group, *CLOCK* expression did not differ between 0700 h and 1900 h (*p* = 0.1465) (Figure 3b). In contrast, the expression of *BMAL1* was significantly higher (*p* < 0.0001) and *CRY1* was significantly lower (*p* = 0.0043) at 1900 h compared with 0700 h (Figure 3b). Here, we observe an interesting inversion of gene expression patterns between DW and NW groups (0700 h vs. 1900 h). Finally, the *PER1* expression was increased at 1900 h (*p* = 0.0001) (Figure 3b).

Given that one of the aims of this study is to compare CLOCK genes’ expression between DW and NW, we seek to understand the genetic patterns through samples taken at 0700 h and 1900 h separately. We found that *CLOCK* expression at 0700 h was increased in NW compared to DW (*p* < 0.0001) (Figure 4a). *BMAL1* showed no differences between groups (*p* = 0.7698) (Figure 4a). Notably, CRY1 expression was increased in NW compared to DW at 0700 h (*p* < 0.0001) (Figure 4a). *PER1* expression did not differ between groups (*p* = 0.6278) (Figure 4a).

However, when we analyzed the gene expression of samples collected at 1900 h, we noticed the maintenance of increased *CLOCK* expression (*p* < 0.0001), which was associated with a marked increased of *BMAL1* expression (*p* < 0.0001) in NW (Figure 4b). Surprisingly, *CRY1* expression was reduced in NW compared with DW (*p* < 0.0001) (Figure 4b), showing a pattern of gene expression inverse to that shown in samples collected at 0700 h. Finally, *PER1* expression increased in the NW group (*p* = 0.0034) (Figure 4b).

### 3.3. Endoplasmic Reticulum Stress

To identify the potential effect of chronic night shift work on genes related to the oxidative process and ERS, we sought to identify the gene expression patterns of PBMC samples from both groups collected at 0700 h and 1900 h. Analyses of samples collected at 0700 h indicated that the NW group showed a robust reduction in *NRF2* mRNA levels (*p* = 0.0015), which is an important regulator of the adaptive response to oxidative stress (Figure 5a). In contrast, the gene expression of *EIF2AK3* (*p* < 0.0001), *ATF4* (*p* < 0.0001), *ATF6* (*p* < 0.0001), and *GRP78* (*p* = 0.0001) was increased in NW compared with DW, while the expression of *CHOP* (*p* = 0.1470) and *XBP1* (*p* = 0.5262) have not been changed (Figure 5a). Consistent with the previous findings, we identified the same patterns of gene alterations in samples collected at 1900 h. Again, NW had a decrease in *NRF2* mRNA level (*p* < 0.0001); however, *EIF2AK3* (*p* < 0.0001), *ATF4* (*p* < 0.0001), *ATF6* (*p* < 0.0001), and *GRP78* (*p* = 0.0001) mRNA levels were increased in NW compared with DW (Figure 5b). Lastly, *CHOP* (*p* = 0.8754) and *XBP1* (*p* = 0.8771) mRNA levels remained unchanged (Figure 5b). The gene expression data normalized by the DW 07:00 h are shown in Appendix A.

Analysis of gender difference in ERS genes expression indicated that women in the NW group are more sensitive to the reduction of *NRF2* mRNA expression than men (NW women *p* = 0.0014 vs. NW men *p* = 0.0398). However, they show a greater increase in gene expression of the *ATF4* (NW women *p* = 0.0026 vs. NW men *p* = 0.0044) and *ATF6* (NW women *p* < 0.0001 vs. NW men *p* = 0.0048), compared to NW men (Appendix A). In addition, the samples collected at 1900 h showed that the effects of night work are more evident in women in the NW group, mainly on *NRF2* (NW women *p* < 0.0001 vs. NW men *p* = 0.2368); however, there are no differences between gender concerning the ERS genes (Appendix A).

We perform correlation studies to assess whether changes in ERS gene expression correlate with circadian clock gene regulation. Notably, we found significant positive correlation between gene expression of *EIF2AK3* (r = 0.663, *p* = 0.002) and *ATF4* (r = 0.587, *p* = 0.008) with *CLOCK* genes at 0700 h in the NW group (Appendix A). In samples collected at 1900 h, we noticed a strong positive correlation between *EIF2AK3* and *CLOCK* (r = 0.724, *p* < 0.001) (Appendix A). This result supports the causality between the circadian clock genes and the alterations of ERS genes in individuals who work the night shift.

## 4. Discussion

In this study, we found that chronic night work in hospital workers directly affects metabolism and increases the risk of developing metabolic syndrome in individuals who work the night shift. Sleep pattern changes promoted by night work result in circadian cycle misalignment and ERS activation. The general characteristics of the group of workers evaluated in this study, as shown in Table 1, added to the fact that they were in the same work environment for at least 5 years, performing specific and coordinated functions, which enabled demonstrating these conclusions. Even for participants in the NW group who reported that they sleep more hours on working days than the DW group, physiological and molecular changes were found that attest to the deleterious effects of night shift work.

Poor diet, stress, and physical inactivity are the main factors contributing to the development of obesity, T2D, and cardiovascular disease, which are some of the main threats to human health [16]. Here, we found that several metabolites are elevated in individuals who work at night compared to DW, such as fasting glucose and HbA1c. Evidence linking the circadian cycle uncontrolled to diabetes shows that shift workers are at a higher risk of developing type 2 diabetes [21,22]. In addition, some studies showed that the relative risk of presenting diabetes is higher in individuals who worked consecutive night shifts compared to individuals who occupy jobs with traditional schedules [23]. Sleep deprivation or even poor sleep quality are risk factors for the appearance or exacerbation of insulin resistance and may affect appetite and adiposity [24,25]. Studies have reported that sleep continence can affect cognitive and physical performance, impairing metabolic functions, such as altered rhythm of circadian melatonin and affecting growth hormone production and being associated with the development of metabolic syndrome, hypertension, and inflammatory process [25,26]. We did not objectively evaluate the sleep period of the participants, but they reported that during the day, it was usually more fragmented (data not shown), which did affect the NW directly.

In this study, we found that the lipid profile of night workers is increased. Triglycerides and LDL levels of NW are increased compared to DW, and HDL levels are lower in NW compared to DW. Some studies showed associations between night shift work and increased food intake, with a preference for carbohydrate foods and changes in lipid profiles, especially triglyceride levels [26,27]. Our findings are consistent with other studies showing that night workers have higher plasma LDL levels than day workers [28].

The anthropometric results showed that the individuals working at night had WC and WHR values increased compared to DW, even though there was no difference in BMI between the groups. These data suggest a central redistribution of body fat, which has been related to an increased risk of insulin resistance and metabolic syndrome [29]. Previous authors have shown that this issue can be explained due to frequent snacks during the work period, reduced duration or absence of sleep time, and exposure to intense white light during the night of work modulating the reduced sensitivity of the internal and external rhythms [11,30]. Although much evidence shows the great impact of the association between night work and obesity, the mechanisms responsible for the connection between these factors are still unclear [31,32].

Another important finding of this study is that NW have systemic arterial pressure, systolic and diastolic, increased compared with DW. Epidemiological studies describe that shift work has negative effects on worker health, which is possibly due to its impact on sleep–wake cycles, eating habits, thermogenesis, and blood pressure levels [33,34,35]. The risk of systemic arterial hypertension (S.A.H.) in individuals with reduced sleep duration is significantly increased as is the risk for obesity and diabetes. This finding is consistent with the hypothesis that both the increase in adipose tissue and insulin resistance can act as mediators in the relationship between reduced sleep time and hypertension. In this way, sleep deprivation in healthy individuals may increase the risk of hypertension and the activation of the sympathetic nervous system [36].

One approach to determine the endogenous circadian rhythm is to assess plasma cortisol concentration. Cortisol, the end-effector of the hypothalamic–pituitary–adrenal axis, is related to anti-inflammatory responses, gluconeogenesis, and immunosuppressors [37,38,39] and function as a humoral signal from the central nervous system (C.N.S.) to reset peripheral clock gene expression [40]. The C.N.S. is the main synchronizer of the human circadian rhythm and coordinates the circadian controllers in the brain and peripheral tissues through signals generated in the suprachiasmatic nucleus of the hypothalamus [33].

Interestingly, we did not observe differences in cortisol concentration between NW and DW in samples collected at 0700 h and 1900 h. Cortisol levels are elevated at 0700 h, and at 1900 h, they were reduced in all the individuals evaluated, preserving their normal physiologic pattern of oscillation. Thus, although limited, our data suggest that there is a conservation of the central circadian pacemakers of night workers even with the chronic exposure of light during the nocturnal work period. In contrast with other studies, increased cortisol levels have been found in individuals with chronic stress, including night work and changes in eating patterns, leading to the onset of metabolic syndrome [11,30].

The misalignment of the circadian cycle promoted by night work has been considered a great contributor to weight gain and visceral obesity [41,42].

Our results suggests that peripheral circadian clocks in NW are misaligned. We identified that the peripheral *CLOCK* expression pattern is not altered in NW. However, when we evaluated *BMAL1* gene expression comparing DW with NW, we noticed the inversion of gene expression in NW, which was possibly induced by chronic night work. Another important factor that we observed was the inversion of *CRY1* expression in NW, which was reduced to 1900 h. Furthermore, we found that the expression of *PER1* in NW remained unchanged, following the same pattern as DW. These data suggest that the peripheral circadian clocks’ elements most sensitive to night shift work are *BMAL1* and *CRY1*. Our data agree with a study that showed that nurses and midwives with *CRY1* alterations were higher in the night shift group than day workers [43].

Circadian misalignment of NW may correlate directly with metabolic disturbances observed in night workers’ clinical data compared to day workers. Skene et al. showed the link between prolonged work shift exposure and the spectrum of metabolic disturbances due to the deregulation of peripheral oscillators [44]. Another study demonstrated the importance of BMAL1 in coordinating insulin secretion with the sleep–wake cycle and how BMAL1 ablation can trigger the onset of diabetes mellitus [45].

Investigating the samples collected at 0700 h, we noticed a different *CLOCK* and *CRY1* expression pattern, and those two genes are increased in NW compared to DW. We also observed alterations in the samples collected at 1900 h, where we showed that the *CLOCK* and *BMAL1* genes remain high in NW; however, we see that *CRY1* gene expression is reduced in NW compared to DW. The circadian clock is formed by feedback cycles with positive and negative components. Thus, CLOCK and BMAL1 are the positive components that are increased by day, while PER1 and CRY1 are negative components of the feedback action that are increased at night [6,46,47,48]. Here, we observe the deregulation of peripheral sensors in night workers, mainly in the expression of *CLOCK* and *BMAL1* that remain elevated even at night, which is possibly due to the chronic stimulation of artificial light from the night work environment.

Interestingly, we report a down-regulation of *NRF2* mRNA expression in NW compared to DW *NRF2*, which is an important oxidative stress regulator in many cell types [49]. This is the first study demonstrating that night work correlates with *NRF2* mRNA expression in peripheral blood samples from NW. Our findings suggest that night shift work can alter the physiological response to oxidative stress. However, further studies should be carried out to validate this hypothesis.

One of the main results of this study is the discovery of ERS activation in NW. Clinical studies showed that ERS has a major impact on metabolic syndromes, including obesity, diabetes, and myocardial dysfunction [50]. In vivo and in vitro recent studies have reported the important role of ATF4 in the circadian regulation, showing that the activation of ER stress is able to inhibit the transcription of the circadian clock and of the clock-controlled genes through an ATF4-dependent mechanism [51,52]. We identify that gene expression of *EIF2AK3*, *ATF4*, *ATF6,* and *GRP78,* which are key elements of ERS activation, are up-regulated in NW. Our findings are in agreement with a study showing that ER stress activation impairs the expression of the circadian clock mainly by an ATF4-dependent mechanism [52]; in addition, an animal study has shown that chronic sleep fragmentation promotes increased food intake and body weight mainly mediated by the activation of ERS [53]. 

Thus, our data show that night shift work can decrease the gene expression of an important marker of resistance to oxidative stress, which is a relevant factor associated with the aging process [54]. In addition, NW presented an ERS activation, which several studies have demonstrated is related to an increased risk of developing metabolic syndrome [16]. Interesting study by Wible et al. demonstrated that the loss of NRF2 function in some types of cells such as fibroblasts and hepatocytes was able to change circadian rhythms in addition to significantly reducing the duration of the cell circadian period, demonstrating in a way that NRF2 is required for normal circadian timing [55]. This study reinforces our hypothesis that night work misaligns the circadian cycle by several metabolic pathways; however, more complete future studies must be performed to prove our hypothesis. Our findings show that women who work at night shift are more sensitive to changes in ERS gene expression than men. However, more in-depth studies may shed light on the underlying mechanisms of this difference between genders mainly because in this study, we did not address gender-specific confusions such as menstrual period, use of contraceptives, and hormonal variations.

Additionally, we also reported a strong correlation between the circadian clock genes and the increase of ERS gene expression in the NW group. Our findings are consistent and support the causality between changes in gene expression of circadian clock genes, ERS, and metabolic disorders.

## 5. Conclusions

In conclusion, our findings open new perspectives for the understanding of mechanisms of how night shift work increases the risk of developing metabolic syndrome, as well as obesity, insulin resistance, and dyslipidemia. In this study, we provide evidence that night workers have significant changes in the expression of peripheral circadian clock genes and activation of ERS-related genes. Our findings add important information to understanding the deleterious effects of night shift work and open a new perspective for working policies with well-designed strategies to reduce stress in the work environment and minimize the metabolic problems arising from night shift work.

## Figures and Tables

**Figure 1 biology-10-00197-f001:**
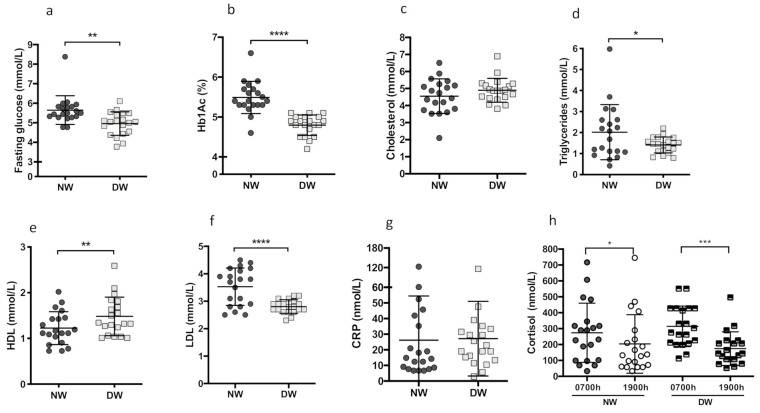
Metabolic changes between day and night workers. Twenty day hospital workers and twenty night hospital workers were evaluated for their metabolic parameters. (**a**) Fasting glucose levels (*p* = 0.0024), (**b**) Hb1Ac (*p* < 0.0001), (**c**) Cholesterol (*p* = 0.2153), (**d**) Triglycerides (*p* = 0.0359), (**e**) HDL-cholesterol (*p* = 0.0456), (**f**) LDL-cholesterol (*p* < 0.0001), (**g**) C Reactive Protein (*p* = 0.9099). Cortisol level was determined in two different blood collections at the beginning of work and another at the end of work. (**h**) Cortisol level, night worker 0700 h vs. 1900 h (*p* = 0.0395), day worker 0700 h vs. 1900 h (*p* = 0.0001). Data are presented as dot plot with mean. Unpaired, one-tailed *t*-test was performed in figures a–g. One-way ANOVA followed by Tukey’s post hoc test, was performed in figure h. * *p* < 0.05; ** *p* < 0.01; *** *p* = 0.001; **** *p* < 0.0001.

**Figure 2 biology-10-00197-f002:**
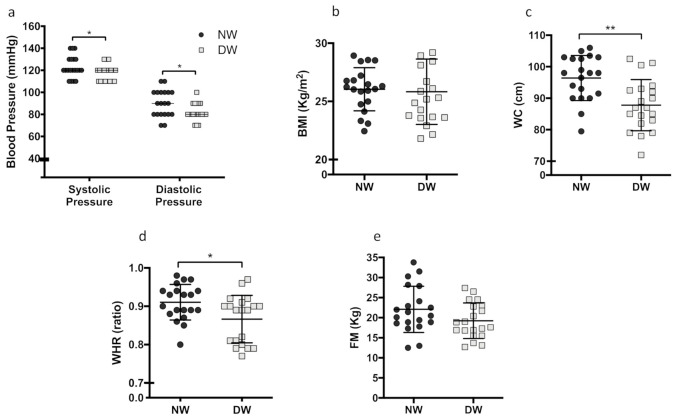
Blood pressure analysis and anthropometric parameters at day and night workers. (**a**) The blood pressure of day and night workers was verified after 20 min of total rest. Systolic pressure (*p* = 0.0496), Diastolic pressure (*p* = 0.0172), (**b**) BMI assessment (*p* = 0.7801), (**c**) Measurement of waist circumference (*p* = 0.0010), (**d**) Waist–hip ratio (*p* = 0.0153), (**e**) Quantification of fat mass (*p* = 0.0887). Data are presented as box plot or dot plot with mean. Unpaired, one-tailed *t*-test. * *p* < 0.05, ** *p* < 0.01. (*n* = 20 per group).

**Figure 3 biology-10-00197-f003:**
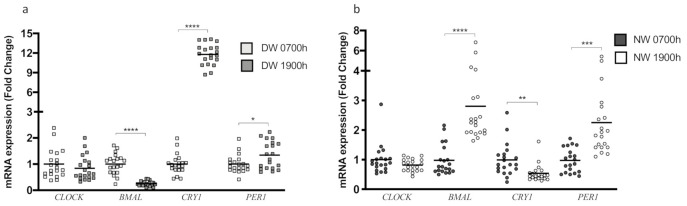
Determination of the gene expression of clock genes in peripheral blood mononuclear cells (PBMC) samples of day and night workers at different times. Gene expression was evaluated by RT-qPCR (*n* = 20 per group). (**a**) Gene expression of the clock genes of day workers at 0700 h and 1900 h. (**b**) Gene expression of the clock genes of night workers at 0700 h and 1900 h. Data are presented as dot plot with mean. Unpaired, one-tailed *t*-test. Data were normalized for values at 0700 h. * *p* < 0.05, ** *p* < 0.01, *** *p* = 0.001, **** *p* < 0.0001.

**Figure 4 biology-10-00197-f004:**
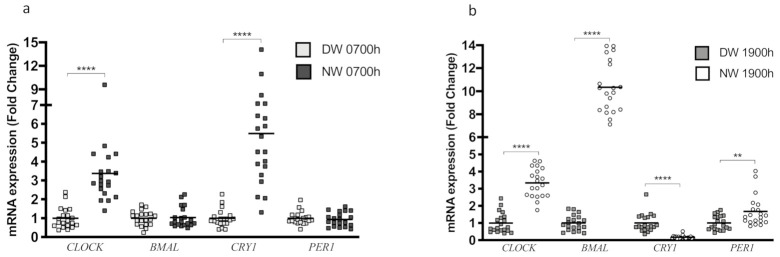
Determination of the gene expression of clock genes in PBMC samples of day and night workers. Gene expression was evaluated by RT-qPCR (*n* = 20 per group). (**a**) Gene expression of the clock genes at 0700 h compared DW and NW. (**b**) Gene expression of the clock genes at 1900 h compared DW and NW. Data are presented as dot plot with mean. Unpaired, one-tailed *t*-test. Data were normalized for DW. ** *p* < 0.01, **** *p* < 0.0001. (*n* = 20 per group).

**Figure 5 biology-10-00197-f005:**
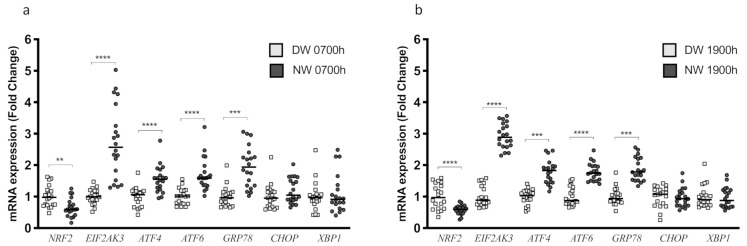
Determination of the expression of genes related to endoplasmic reticulum stress in PBMC samples of day and night workers at 0700 h and 1900 h. (**a**) Graph representing diurnal variations in gene expression of genes related to endoplasmic reticulum stress (ERS). The data were normalized by day workers (DW) 0700 h. (**b**) Graph representing the nocturnal variations of gene expression of genes related to ERS. The data were normalized by DW 1900 h. Gene expression was evaluated by RT-qPCR (*n* = 20 per group). Data are presented as dot plot with mean. Unpaired, one-tailed *t*-test. ** *p* < 0.01, *** *p* = 0.001, **** *p* < 0.0001.

**Table 1 biology-10-00197-t001:** Baseline characteristics of the subjects.

Characteristics	Day Workers (*n* = 20)	Night Workers (*n* = 20)
Age, y	38 (6.8)	40 (4.9)
Sex, % male	10 (50%)	10 (50%)
Weight, kg	70.1 (12.4)	77.3 (10.0)
Height, m	1.64 (0.07)	1.72 (0.08)
Sleep time work day, h	8.1 (0.5)	9.1 (0.6)
Sleep time free day, h	8.5 (0.5)	8.2 (0.4)

Data are presented as means (standard deviation) or percentages. Abbreviations: BMI, body mass index.

## Data Availability

The data presented in this study are available on request from the corresponding author.

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
