# Peer review of "Circadian Misalignment Induced by Chronic Night Shift Work Promotes Endoplasmic Reticulum Stress Activation Impacting Directly on Human Metabolism"

_biology, 2021, doi:10.3390/biology10030197_

Round 1

Reviewer 1 Report

While the authors have addressed some of this reviewer's initial concerns, there are still some issues with data presentation and overstatement of results in the discussion that must be addressed.  

1.  The authors have not amended the unsupported statements that the NW are sleep deprived.  The data are not present to support this idea.  Any suggestion that the data in this manuscript are indicative of sleep deprivation should be removed (i.e., line 38, 74, 292, 333, 373, and 377) or sufficient data should be provided.  

2.  Why are the blood pressure data presented as box plots vs. scatter like the rest of the paper?  The data should be consistent, especially considering how close the comparison appears.  If there are outliers in these data that skew towards statistical significance, the authors should revisit the data (all of the data) and perform outlier analysis, if not done already.

3. Methods section 2.6 should be combined with 2.4 instead of being in its own section.

4.  How were ERS genes normalized?  The use of secondary normalization on qPCR data  should not be necessary.  If normalizing to both GAPDH and 18S rRNA (the former of which may be questionable), that should be sufficient.  As it stands, the "Fold Change" label must be clarified: is this Fold Change over housekeeping genes or in relation to the secondary normalization?

5.  The authors use comparison of p-values to suggest that women are more sensitive to NRF2 reduction than males.  Without effect size analysis, comparing p-values may not be appropriate.  Additionally, the timing of menstruation does not appear to be taken into account, which may have affected the data . Also, without gender-specific data from other tests (blood pressure, morphometrics, etc) these data are not relevant to the central hypothesis.

The authors give no discussion regarding clock control of ERS genes and vice versa.  It has been shown that BMAL:CLOCK activates ATF4, which feeds forward to the PER:CRY complex, yet these relationships are not apparent in the data presented here.  Whether or not this suggests a misaligned clock is worthy of discussion.

There are several places in the discussion where the authors overstate the implications of the data.  Regarding the misalignment of the molecular clock, it is difficult to know the state of the clock when 1) Chronotypes of subjects is not known as well as any potential genetic background that might affect gene expression, 2) no data on Per2, Cry2, Ror-a, or Reverb-a are provided, and 3) the only marker of circadian phase used was cortisol which, in other studies of sleep deprivation, showed significant differences at other times of day (no difference at the same clock-times shown here).  The central clock may in fact be perturbed, but the authors do not have sufficient data to say that it is not.

The conflicting anthropometric data are explained away with hypotheticals such as snacks eaten during work hours, but there is no data provided to support these.  It is surprising that after 5 years of working night shifts, some metrics like blood glucose and HbA1c levels are different, while BMI and blood pressure are not.  These data should be discussed in a broader context, and not cherry-picked to support the hypothesis.

There are still several grammatical errors, the most notable for this reviewer is the lack of periods at the end of sentences (mostly when a sentence ends in "NW" or "DW").  

Author Response

January 15th, 2021

Dr. Chris O'Callaghan

Editor-in-Chief of Biology 

Dr. Eléonore Maury

Special Issue Editor of Biology

Dear Dr. O'Callaghan and Dr. Maury,

I am sending the revised manuscript (biology-1008617) titled, “Circadian misalignment induced by chronic night shift work promotes endoplasmic reticulum stress activation impacting directly on human metabolism” based on the reviewers’ comments. We have addressed all the comments and taken the suggestions of our reviewers. It is important to highlight that we recognize the limitations in our study and we acknowledge it in the manuscript but we believe our data is interesting and improves the knowledge in a very exciting topic in human regulation. 

Attached is the rebuttal letter, the manuscript will hopefully demonstrate adequate improvement to be worthy of publication in Biology.

Thank you very much for your attention.

Rafael Ferraz-Bannitz, MS.

Ribeirao Preto Medical School

University of São Paulo

Maria Cristina Foss Freitas, MD, Ph.D.

Ribeirao Preto Medical School

University of São Paulo

Sincerely,

Response to the reviewers’ comments and suggestions

Reviewer reports:

Reviewer #1

 Comments and Suggestions for Authors

While the authors have addressed some of this reviewer's initial concerns, there are still some issues with data presentation and overstatement of results in the discussion that must be addressed.  

  1. The authors have not amended the unsupported statements that the NW are sleep deprived.  The data are not present to support this idea.  Any suggestion that the data in this manuscript are indicative of sleep deprivation should be removed (i.e., line 38, 74, 292, 333, 373, and 377) or sufficient data should be provided.  

We appreciate the reviewer's observation, and we remove all sleep deprivation statements in the manuscript.

  1. Why are the blood pressure data presented as box plots vs. scatter like the rest of the paper?  The data should be consistent, especially considering how close the comparison appears.  If there are outliers in these data that skew towards statistical significance, the authors should revisit the data (all of the data) and perform outlier analysis, if not done already.

We thank the reviewer's suggestion and reformulated the blood pressure graph leaving it in dot plot format to make the results and graphs more consistent.

  1. Methods section 2.6 should be combined with 2.4 instead of being in its own section.

We followed the reviewer's suggestion and made the changes.

  1. How were ERS genes normalized?  The use of secondary normalization on qPCR data should not be necessary.  If normalizing to both GAPDH and 18S rRNA (the former of which may be questionable), that should be sufficient.  As it stands, the "Fold Change" label must be clarified: is this Fold Change over housekeeping genes or in relation to the secondary normalization?

We use the GAPDH gene as a housekeeping gene to normalize gene expressions. Therefore, the graphs are showing the expression of the ERS-related genes normalized by GAPDH. We insert this information in methods (Line 115-116).

  1. The authors use comparison of p-values to suggest that women are more sensitive to NRF2 reduction than males.  Without effect size analysis, comparing p-values may not be appropriate.  Additionally, the timing of menstruation does not appear to be taken into account, which may have affected the data. Also, without gender-specific data from other tests (blood pressure, morphometrics, etc) these data are not relevant to the central hypothesis.

We used the p-value as thresholds because this parameter was used in all other analyzes we are showing. Important to mention that we also did gender analysis on all tests, but we didn't see any differences. The timing of menstruation is not reported and is a limitation of this study. However, this difference highlights interesting findings that should be further explored to address sleeping behavior changes in different metabolic situations according to gender. The limitation on gender differences is stated in lines 367 and 368.

The authors give no discussion regarding clock control of ERS genes and vice versa.  It has been shown that BMAL:CLOCK activates ATF4, which feeds forward to the PER:CRY complex, yet these relationships are not apparent in the data presented here.  Whether or not this suggests a misaligned clock is worthy of discussion.

We added information of the in vitro and in vivo data on ATF4 role on circadian regulation (lines 359-360)

There are several places in the discussion where the authors overstate the implications of the data.  Regarding the misalignment of the molecular clock, it is difficult to know the state of the clock when 1) Chronotypes of subjects is not known as well as any potential genetic background that might affect gene expression, 2) no data on Per2, Cry2, Ror-a, or Reverb-a are provided, and 3) the only marker of circadian phase used was cortisol which, in other studies of sleep deprivation, showed significant differences at other times of day (no difference at the same clock-times shown here).  The central clock may in fact be perturbed, but the authors do not have sufficient data to say that it is not.

Besides the fact that we did not intend to clarify all aspects of the circadian misalignment in this manuscript, our data shows an interesting correlation of limited but important genes in CLOCK regulation and NRF2 gene expression, an important regulator of oxidative stress. These findings are novel and open new research opportunities, making it important to share with the scientific community. All points listed will be addressed in further research. We agree that the regulation and interaction on peripheral and central clocks should be better addressed and deeper understood in this population.   

The conflicting anthropometric data are explained away with hypotheticals such as snacks eaten during work hours, but there is no data provided to support these.  It is surprising that after 5 years of working night shifts, some metrics like blood glucose and HbA1c levels are different, while BMI and blood pressure are not.  These data should be discussed in a broader context, and not cherry-picked to support the hypothesis.

We did not cherry-pick the information to support our hypothesis. The data presented is all that was possible to collect from the patients; we recognize and acknowledge that our study has limitations. We added information stating this limitation in the discussion (line 297).

Our data showed that glucose and A1c were increased in the night shift workers as well as blood pressure (systolic and diastolic measurements), triglycerides and LDL-cholesterol, consistent with a lower HDL-cholesterol. These data are not conflicting and are shown in figures 1 and 2.

The BMI and body weight were not different between the groups, but waist circumference was increased in the night shift workers, consistent with a higher waist-hip ratio. These data suggest a central redistribution of body fat, which has been related to an increased risk of insulin resistance and metabolic syndrome. Although the BMI measurements are wildly used in clinical practice and research studies, there are limitations, especially related to body fat distribution and body composition. Our study did not aim to evaluate body composition either fat distribution in these individuals, but our data brings light to this hypothesis and the role of lipid partitioning in circadian cycle misalignment. Further studies are necessary and should be encouraged to try to explain this observation.       

There are still several grammatical errors, the most notable for this reviewer is the lack of periods at the end of sentences (mostly when a sentence ends in "NW" or "DW").  

Sorry for this typo, the manuscript was reviewed, and the periods were added at the end of the sentences.

Reviewer 2 Report

In this study, the authors performed clinical and molecular analyses on daytime workers and nighttime workers in a hospital, and revealed that nighttime workers show increased levels of glucose, triglycerides, waist circumference and blood pressure compared to daytime workers. The authors also demonstrated that mRNA expression of core clock genes as well as ER stress-related genes is altered in nighttime workers. The manuscript has been improved, but there are still several concerns that need to be addressed adequately before consideration for the possibility of publication.

Major points:

  1. In Figure 5, the authors show the data obtained both at 7:00 and at 19:00. However, information about day-night variations for expression levels are missing, possibly because the data at 7:00 and at 19:00 seem to be normalized independently and presented separately. As I suggested previously, the authors should combine the data with a single normalization (for example, the data obtained from DW 7:00 is set to 1 and other 3 samples (DW 19:00, NW 7:00, and NW 19:00) are shown as relative values to DW 7:00) for each gene to clarify how much day-night variations are. Otherwise, the authors should draw a separate graph showing the day-night variations in expression levels of each gene and describe the method of normalization clearly as in Figure 3. It would be informative to show the day-night variation of ER stress-related gene expression as the authors actually have the data.

  1. As for interpretation of the data, the authors say in the conclusion section, “In this study, we provide evidence that … in addition to a decreased adaptive response to oxidative stress.” (line 374). However, they only show the decrease in the mRNA expression level of NRF2. The authors should also revise this description to avoid overinterpretation, for example as follows “… in addition to decreased expression of NRF2, the master regulator of antioxidative stress responses.”

Minor point:

  1. I think that “circadian cycle genes (CLOCK genes)” (line 20) is not a common phrase to indicate the core circadian clock genes that constitute transcriptional-translational feedback loops. I recommend to use “circadian clock genes” or “core clock genes”, or just “clock genes” without capitalizing.

Author Response

January 15th, 2021

Dr. Chris O'Callaghan

Editor-in-Chief of Biology 

Dr. Eléonore Maury

Special Issue Editor of Biology

Dear Dr. O'Callaghan and Dr. Maury,

I am sending the revised manuscript (biology-1008617) titled, “Circadian misalignment induced by chronic night shift work promotes endoplasmic reticulum stress activation impacting directly on human metabolism” based on the reviewers’ comments. We have addressed all the comments and taken the suggestions of our reviewers. It is important to highlight that we recognize the limitations in our study and we acknowledge it in the manuscript but we believe our data is interesting and improves the knowledge in a very exciting topic in human regulation. 

Attached is the rebuttal letter, the manuscript will hopefully demonstrate adequate improvement to be worthy of publication in Biology.

Thank you very much for your attention.

Rafael Ferraz-Bannitz, MS.

Ribeirao Preto Medical School

University of São Paulo

Maria Cristina Foss Freitas, MD, Ph.D.

Ribeirao Preto Medical School

University of São Paulo

Sincerely,

Reviewer #2:

In this study, the authors performed clinical and molecular analyses on daytime workers and nighttime workers in a hospital, and revealed that nighttime workers show increased levels of glucose, triglycerides, waist circumference and blood pressure compared to daytime workers. The authors also demonstrated that mRNA expression of core clock genes as well as ER stress-related genes is altered in nighttime workers. The manuscript has been improved, but there are still several concerns that need to be addressed adequately before consideration for the possibility of publication.

Major points:

  1. In Figure 5, the authors show the data obtained both at 7:00 and at 19:00. However, information about day-night variations for expression levels are missing, possibly because the data at 7:00 and at 19:00 seem to be normalized independently and presented separately. As I suggested previously, the authors should combine the data with a single normalization (for example, the data obtained from DW 7:00 is set to 1 and other 3 samples (DW 19:00, NW 7:00, and NW 19:00) are shown as relative values to DW 7:00) for each gene to clarify how much day-night variations are. Otherwise, the authors should draw a separate graph showing the day-night variations in expression levels of each gene and describe the method of normalization clearly as in Figure 3. It would be informative to show the day-night variation of ER stress-related gene expression as the authors actually have the data.

 We followed the reviewer's suggestion and made a graph showing the genetic expression of all groups normalized by DW 07: 00h. Supplementary Figure. We also modified the legend using the reviewer's suggestion to make it clearer to the reader how the qPCR data in each figure was normalized.

  1. As for interpretation of the data, the authors say in the conclusion section, “In this study, we provide evidence that … in addition to a decreased adaptive response to oxidative stress.” (line 374). However, they only show the decrease in the mRNA expression level of NRF2. The authors should also revise this description to avoid overinterpretation, for example as follows “… in addition to decreased expression of NRF2, the master regulator of antioxidative stress responses.”

 We thank and agree with the reviewer. We only observed a decrease in the expression of NRF2 mRNA, so we cannot overestimate this data. We removed this information in the completion section (Line 374-375)

Minor point:

  1. I think that “circadian cycle genes (CLOCK genes)” (line 20) is not a common phrase to indicate the core circadian clock genes that constitute transcriptional-translational feedback loops. I recommend to use “circadian clock genes” or “core clock genes”, or just “clock genes” without capitalizing

We thank the reviewer's attentive view and modify the expression on line 20 as suggested

Reviewer 3 Report

It reads much better as my grad students and I agree. Thank you for addressing the vast majority of our concerns. We appreciate the opportunity to review articles written by fellow circadian rhythms researchers. 

Author Response

January 15th, 2021

Dr. Chris O'Callaghan

Editor-in-Chief of Biology 

Dr. Eléonore Maury

Special Issue Editor of Biology

Dear Dr. O'Callaghan and Dr. Maury,

I am sending the revised manuscript (biology-1008617) titled, “Circadian misalignment induced by chronic night shift work promotes endoplasmic reticulum stress activation impacting directly on human metabolism” based on the reviewers’ comments. We have addressed all the comments and taken the suggestions of our reviewers. It is important to highlight that we recognize the limitations in our study and we acknowledge it in the manuscript but we believe our data is interesting and improves the knowledge in a very exciting topic in human regulation. 

Attached is the rebuttal letter, the manuscript will hopefully demonstrate adequate improvement to be worthy of publication in Biology.

Thank you very much for your attention.

Rafael Ferraz-Bannitz, MS.

Ribeirao Preto Medical School

University of São Paulo

Maria Cristina Foss Freitas, MD, Ph.D.

Ribeirao Preto Medical School

University of São Paulo

Sincerely,

Reviewer #3

It reads much better as my grad students and I agree. Thank you for addressing the vast majority of our concerns. We appreciate the opportunity to review articles written by fellow circadian rhythms researchers. 

We appreciate the reviewer's commitment and attention helping to improve the quality of our manuscript

Round 2

Reviewer 1 Report

This reviewer would like to thank the authors for the revised manuscript; it is in much better shape.  However, some of the edits made do not address the underlying problem with the quality of discussion included.  Overall, the results are interesting, but obscured by the limitations of the experiment.  This reviewer understands that sometimes limitations are unavoidable, but the authors have missed several opportunities to put some of their results into a biological/physiological context.

Please refer to the submitted response letter for the below issues

4.  The methods section is now less comprehensible.  Section 2.4 states that GAPDH was used as a housekeeping gene, but section 2.5 says both GAPDH and 18s rRNA were used.  The quantification must be detailed, by experiment if need be, and a justification must be provided if two different normalization methods were used for different experiments.  Otherwise, the data from one set of qPCR experiments cannot be compared to another that was analyzed differently.

5. The authors have failed to indicate that gender-specific confounds, such as menstruation, were not addressed in these data.  The edited lines that were referenced do not allude to this either (367-368), only that future study is needed (369-371).  

The authors completely missed the chance to dicsuss the role of ATF4 in clock function here.  The provided statement:

"In vivo and in vitro recent studies have reported the important role of ATF4 in the circadian regulation, showing that the activation of ER stress is able to inhibit the transcription of the circadian clock and of the clock-controlled genes through an ATF4-dependent mechanism."

does NOT address the authors' own data.  If the data shown are in agreement here, state that.  If the data are in conflict with what we know about clock regulation of ERS, what could that mean in the context of an "intact central pacemaker" and disrupted peripheral clock mechanism?

Similarly, NRF2 has been shown to directly affect circadian rhythms (Wible et al, eLife, 2018).  Could it be that shiftwork affects ERS, which would alter NRF2 expression, which would then feed into peripheral clocks?  Is it the other way around?  The authors should include discussion on this.

This reviewer agrees with the idea that differences in anthropometric data could suggest a redistribution of fat, so the authors should include this in the discussion.  Instead the provided statement in line 297:

"Although we did not evaluate show data from our group, previous authors have shown that..." is obviously a typo and makes no sense.

Author Response

January 24th, 2021

Dr. Chris O'Callaghan

Editor-in-Chief of Biology 

Dr. Eléonore Maury

Special Issue Editor of Biology

Dear Dr. O'Callaghan and Dr. Maury,

I am sending the second revised manuscript (biology-1008617) titled, “Circadian misalignment induced by chronic night shift work promotes endoplasmic reticulum stress activation impacting directly on human metabolism” based on the reviewer comments. We have addressed all the comments and taken the suggestions of our reviewers. It is important to highlight that we recognize the limitations in our study and we acknowledge it in the manuscript but we believe our data is interesting and improves the knowledge in a very exciting topic in human regulation. 

Attached is the second rebuttal letter, the manuscript will hopefully demonstrate adequate improvement to be worthy of publication in Biology.

Thank you very much for your attention.

Rafael Ferraz-Bannitz, MS.

Ribeirao Preto Medical School

University of São Paulo

Maria Cristina Foss Freitas, MD, Ph.D.

Ribeirao Preto Medical School

University of São Paulo

Sincerely,

Response to the reviewer comments and suggestions

1# Reviewer

This reviewer would like to thank the authors for the revised manuscript; it is in much better shape.  However, some of the edits made do not address the underlying problem with the quality of discussion included.  Overall, the results are interesting, but obscured by the limitations of the experiment.  This reviewer understands that sometimes limitations are unavoidable, but the authors have missed several opportunities to put some of their results into a biological/physiological context.

Please refer to the submitted response letter for the below issues

  1. The methods section is now less comprehensible.  Section 2.4 states that GAPDH was used as a housekeeping gene, but section 2.5 says both GAPDH and 18s rRNA were used.  The quantification must be detailed, by experiment if need be, and a justification must be provided if two different normalization methods were used for different experiments.  Otherwise, the data from one set of qPCR experiments cannot be compared to another that was analyzed differently.

We used the GAPDH gene as a housekepping gene for all gene expression analyzes in this study. We also performed an analysis using the 18s gene, but it was not used to normalize any gene expression data. Agreed with the reviewer and removed ambiguous information from the methods (line 124).

  1. The authors have failed to indicate that gender-specific confounds, such as menstruation, were not addressed in these data.  The edited lines that were referenced do not allude to this either (367-368), only that future study is needed (369-371).  

We agree with the reviewer and therefore indicated the limitations of these interpretations on line 370-373.

The authors completely missed the chance to dicsuss the role of ATF4 in clock function here.  The provided statement:

"In vivo and in vitro recent studies have reported the important role of ATF4 in the circadian regulation, showing that the activation of ER stress is able to inhibit the transcription of the circadian clock and of the clock-controlled genes through an ATF4-dependent mechanism."

does NOT address the authors' own data.  If the data shown are in agreement here, state that.  If the data are in conflict with what we know about clock regulation of ERS, what could that mean in the context of an "intact central pacemaker" and disrupted peripheral clock mechanism?

We improved the discussion on the effect of ERS on the circadian cycle and included our data in this paragraph. (line 363-367).

Similarly, NRF2 has been shown to directly affect circadian rhythms (Wible et al, eLife, 2018).  Could it be that shiftwork affects ERS, which would alter NRF2 expression, which would then feed into peripheral clocks?  Is it the other way around?  The authors should include discussion on this.

We include this article in the discussion and parallel the results of this article with our results

This reviewer agrees with the idea that differences in anthropometric data could suggest a redistribution of fat, so the authors should include this in the discussion.  Instead the provided statement in line 297:

We include this observation in the discussion (Line 297-298)

"Although we did not evaluate show data from our group, previous authors have shown that..." is obviously a typo and makes no sense.

We fixed this error, we appreciate the reviewer's attention

This manuscript is a resubmission of an earlier submission. The following is a list of the peer review reports and author responses from that submission.

Round 1

Reviewer 1 Report

Ferraz-Bannits and colleagues present clinical, anthropometric, and molecular data on 40 hospital works split evenly between those working 5+ years as Night Workers and 6+ years as Day Workers.  Their stated hypothesis is that night work induces metabolic disorders such as increased fasting glucose, Hb1Ac, triglycerides, and LDL.  They also state that they identify that chronic sleep deprivation induced by night work affects peripheral clock genes and induces ERS activation in night workers.  The latter point is presented as data from peripheral blood mononuclear cells from two time points.  

While the paper is mostly written well, there are several grammatical errors that must be addressed. However, there are more serious concerns regarding the data and the authors' interpretation.

The sleep data, which are self-reported by the study participants, does not suggest sleep deprivation within night workers.  Similarly, it is difficult to put the only measure of sleep (hours of daily sleep), without having metrics of participant chronotype.  As it is, the suggestion that the molecular data might be borne out of less sleep is not supported. 

The clock gene expression data would be more indicative of clock function or strength if the reader had some idea of the phase of the participants.  This is especially true considering only two samples were taken.  It is likely that the sampling times occurred adjacent to peaks and troughs of expression, leading to damped amplitude when the time points are compared to each other. At the very least, melatonin should have been sampled to establish some context to the central pacemaker.

In figure 1, the authors state that there is higher variability in cortisol among night workers, but this is not tested.  The authors should've included either CV or used median levels to support this. 

Figure 3 and 4 include some samples with very large errors.  These figures should include scatter plots for each gene to show that means have not been skewed by outliers.

More concerning is the differences in gene expression between these two figures.  Clock and Cry1 in Fig4a are very different in value compared to Figure 3 and Clock and Bmal1 in Figure 4b are also different. This would suggest that the authors did some normalization after the delta-delta-CT method explained in the methods.  These are major discrepancies and must be addressed.

Finally, the authors show ERS-related gene expression data and state that increased levels in these genes indicate night work causes ERS that leads to metabolic syndrome.  However, the authors specifically state that GRP78 is increased when it clearly is decreased in both Figure 5 and the supplemental figure.  How can the authors state that ERS is increased when a major gene known to have a direct relationship with ERS is down-regulated?  

While there is novelty in investigating ERS as it pertains to the clock, much of the data shown here has already been shown, and with clearer results.  This study would greatly benefit from higher sampling resolution, inclusion of melatonin levels, and clearer representation of gene expression data.

Reviewer 2 Report

In this study, the authors performed clinical and molecular analyses on daytime workers and nighttime workers in a hospital, and revealed that nighttime workers show increased levels of glucose, triglycerides, waist circumference and blood pressure compared to daytime workers. The authors also demonstrated that mRNA expression of core clock genes as well as ER stress-related genes is altered in nighttime workers. The aim and design of the study is sound and adequate. However, there are several concerns mainly about interpretation of data that need to be addressed adequately before consideration for the possibility of publication.

Major points:

  1. It seems that Figure 3 and 4 show the result of the same dataset. The authors should combine these Figures by showing fold changes relative to the sample obtained from daytime workers at 7:00 to clarify the difference between daytime workers and nighttime workers with day-night changes of expression levels.

  1. In Figure 5, the authors show the data from only the samples obtained at 7:00 for both daytime and nighttime workers. However, the difference observed in expression levels between daytime and nighttime workers may simply reflect the diurnal variations in accordance with the internal circadian rhythms. I recommend that the authors also show the data obtained at 19:00, which would provide further information about apparent changes in ER stress-related gene expression.

  1. The authors say, “One of the main results of this study is the discovery of decreased oxidative stress response” (line 323) and “our data show that NW have decreased protection against oxidative stress” (line 332). However, they only show the decrease in the mRNA expression level of Nrf2. As Nrf2 activity is also regulated by posttranslational mechanisms, it is not clear whether oxidative stress response is actually decreased. The authors should at least show the expression levels of some Nrf2 target genes such as Nqo1, Gsta1, and Gstm1, or avoid overinterpretation of the result.

Minor point

The authors should carefully revise the typos and grammatical errors throughout the manuscript.

Author Response

December 21st, 2020

Dr. Chris O'Callaghan

Editor-in-Chief of Biology 

Dr. Eléonore Maury

Special Issue Editor of Biology

Dear Dr. O'Callaghan and Dr. Maury,

I am sending the revised manuscript (biology-1008617) titled, “Circadian misalignment induced by chronic night shift work promotes endoplasmic reticulum stress activation impacting directly on human metabolism” based on the reviewers’ comments. We have addressed all the comments and taken the suggestions of our reviewers. We believe that our manuscript is improved in light of these comments and suggestions.

Attached is the rebuttal letter, the manuscript will hopefully demonstrate adequate improvement to be worthy of publication in Biology.

Thank you very much for your attention.

Sincerely,

Rafael Ferraz-Bannitz, MS.

Ribeirao Preto Medical School

University of São Paulo

Maria Cristina Foss-Freitas, MD, Ph.D.

Ribeirao Preto Medical School

University of São Paulo

Reviewer #2: In this study, the authors performed clinical and molecular analyses on daytime workers and nighttime workers in a hospital, and revealed that nighttime workers show increased levels of glucose, triglycerides, waist circumference and blood pressure compared to daytime workers. The authors also demonstrated that mRNA expression of core clock genes as well as ER stress-related genes is altered in nighttime workers. The aim and design of the study is sound and adequate. However, there are several concerns mainly about interpretation of data that need to be addressed adequately before consideration for the possibility of publication.

Major points:

  1. It seems that Figure 3 and 4 show the result of the same dataset. The authors should combine these Figures by showing fold changes relative to the sample obtained from daytime workers at 7:00 to clarify the difference between daytime workers and nighttime workers with day-night changes of expression levels.

We are grateful for the reviewer's suggestion. In fact, Figures 3 and 4 show the results of the data on the expression of the CLOCK genes; however, we understand that placing them separately would be a good way to facilitate the readers' interpretation of the data without presenting a figure full of information.

  1. In Figure 5, the authors show the data from only the samples obtained at 7:00 for both daytime and nighttime workers. However, the difference observed in expression levels between daytime and nighttime workers may simply reflect the diurnal variations in accordance with the internal circadian rhythms. I recommend that the authors also show the data obtained at 19:00, which would provide further information about apparent changes in ER stress-related gene expression.

We followed the reviewer's suggestion and inserted the data and the gene expression figure of the ERS-related genes in samples taken from 1900h.

  1. The authors say, “One of the main results of this study is the discovery of decreased oxidative stress response” (line 323) and “our data show that NW have decreased protection against oxidative stress” (line 332). However, they only show the decrease in the mRNA expression level of Nrf2. As Nrf2 activity is also regulated by posttranslational mechanisms, it is not clear whether oxidative stress response is actually decreased. The authors should at least show the expression levels of some Nrf2 target genes such as Nqo1, Gsta1, and Gstm1, or avoid overinterpretation of the result.

We are grateful for the suggestion, and we cannot misinterpret our data. In this way, we change the description and interpretation of the data.

Minor point

The authors should carefully revise the typos and grammatical errors throughout the manuscript.

Thanks to the reviewer and we performed a grammatical revision.

Reviewer 3 Report

Congratulations to the authors on what looks like lots of hard work. I think this article makes an important contribution. However we have suggestions.

Regarding the title of the article, it may be beneficial to write out “endoplasmic reticulum” as opposed to using an acronym. ER in the title may be misleading or easily confused until the summary or abstract has been read.

Overall, the paper has significant grammatical errors as well as word choice problems making the intended meaning unclear. We understand of course and have made suggestions but these are not exhaustive.

The following are a few of the most substantial errors - Instead of self-powered on lines 56-57, it would be better changed to self-regulating.  “Induced by” on line 75 would be better changed to something like “because of” or “as a result of.” Line 261 the statement “according to our findings other studies showed” is poorly constructed and is not clear.  

For participants, were there any exclusion criteria or just inclusive criteria? Line 94 indicates a questionnaire was used to collect data on usual sleep duration, what questionnaire was used? Was the questionnaire standardized or validated? What exactly constitutes “sleep duration” as this can have different potential meanings. We can only assume that subjects were asked to self-report certain parameters measuring their sleep, whether that be total time in bed or total sleep duration, we cannot ascertain. Typically, self-reporting without structure in behavioral research leaves the results open to vulnerabilities and extraneous factors impacting the results. The use of actigraphy would be strongly recommended in any follow-up. But can you at least provide us with what tool you used for the self report of sleep amount?

For statistical analyses, it states that either a t-test or an ANOVA when appropriate. When t-tests or ANOVA were used or were appropriate to use should be specified/indicated. The use of effect sizes would also be recommended.

Lines 167-173 outline the data indicated in figure 2, however, p-values were not indicated in the text, only the figure. The results outlined in lines 167-173 are misleading as only of the variables (waist circumference) is stated to be significant but the p-values reported in figure 2 clearly indicate that systolic pressure, diastolic pressure, and waist-hip ratio are also statistically significant. Only stating that the results were "higher" or "increased" is different from stating that a statistical significance was found. Line 188 indicates that the expression of CRY1 and PERI1 showed an increase at 19:00h. According to figure 3, this increase is of statistical significance. It is confusing why the authors seem almost reticent to state in the text that they found significant differences.

Otherwise, we enjoyed this article. We appreciate the opportunity to review chronobiological sciences.

Author Response

December 21st, 2020

Dr. Chris O'Callaghan

Editor-in-Chief of Biology 

Dr. Eléonore Maury

Special Issue Editor of Biology

Dear Dr. O'Callaghan and Dr. Maury,

I am sending the revised manuscript (biology-1008617) titled, “Circadian misalignment induced by chronic night shift work promotes endoplasmic reticulum stress activation impacting directly on human metabolism” based on the reviewers’ comments. We have addressed all the comments and taken the suggestions of our reviewers. We believe that our manuscript is improved in light of these comments and suggestions.

Attached is the rebuttal letter, the manuscript will hopefully demonstrate adequate improvement to be worthy of publication in Biology.

Thank you very much for your attention.

Sincerely,

Rafael Ferraz-Bannitz, MS.

Ribeirao Preto Medical School

University of São Paulo

Maria Cristina Foss-Freitas, MD, Ph.D.

Ribeirao Preto Medical School

University of São Paulo

Reviewer #3: Congratulations to the authors on what looks like lots of hard work. I think this article makes an important contribution. However we have suggestions.

Regarding the title of the article, it may be beneficial to write out “endoplasmic reticulum” as opposed to using an acronym. ER in the title may be misleading or easily confused until the summary or abstract has been read.

We are grateful for the reviewer's suggestion and made the change to the paper title.

Overall, the paper has significant grammatical errors as well as word choice problems making the intended meaning unclear. We understand of course and have made suggestions but these are not exhaustive.

We apologize for the grammatical errors. The text was revised.

The following are a few of the most substantial errors - Instead of self-powered on lines 56-57, it would be better changed to self-regulating.  “Induced by” on line 75 would be better changed to something like “because of” or “as a result of.” Line 261 the statement “according to our findings other studies showed” is poorly constructed and is not clear.  

We appreciate the reviewer's corrections and made the changes as suggested by the reviewer.

For participants, were there any exclusion criteria or just inclusive criteria? Line 94 indicates a questionnaire was used to collect data on usual sleep duration, what questionnaire was used? Was the questionnaire standardized or validated? What exactly constitutes “sleep duration” as this can have different potential meanings. We can only assume that subjects were asked to self-report certain parameters measuring their sleep, whether that be total time in bed or total sleep duration, we cannot ascertain. Typically, self-reporting without structure in behavioral research leaves the results open to vulnerabilities and extraneous factors impacting the results. The use of actigraphy would be strongly recommended in any follow-up. But can you at least provide us with what tool you used for the self report of sleep amount?

The inclusion criteria were workers without metabolic diseases, cancer or pregnancies, in addition to having been working for at least 4 years in the same work shift. We agree with the reviewer's arguments. However, we use only one non-validated questionnaire with the following questions:

1- How long have I been working at this hospital?

2- What work shift do you belong to? How long have you worked on the same shift?

3- How long do you sleep on a normal work day?

4- How long do you sleep on a day off after your workday?

5- Do you have diseases like cancer, metabolic disease or this pregnant woman?

Based on this, the inability to precisely confirm the data on the participants' sleep period is a limitation of our study.

For statistical analyses, it states that either a t-test or an ANOVA when appropriate. When t-tests or ANOVA were used or were appropriate to use should be specified/indicated. The use of effect sizes would also be recommended.

We made the changes suggested by the reviewer indicating in the legend, which was the statistical methods used in each data analysis.

Lines 167-173 outline the data indicated in figure 2, however, p-values were not indicated in the text, only the figure. The results outlined in lines 167-173 are misleading as only of the variables (waist circumference) is stated to be significant but the p-values reported in figure 2 clearly indicate that systolic pressure, diastolic pressure, and waist-hip ratio are also statistically significant. Only stating that the results were "higher" or "increased" is different from stating that a statistical significance was found. Line 188 indicates that the expression of CRY1 and PERI1 showed an increase at 19:00h. According to figure 3, this increase is of statistical significance. It is confusing why the authors seem almost reticent to state in the text that they found significant differences.

We agree with the reviewer's observations and for that reason, we rewrote the results to make them clearer to the readers, and we also added the p-value in all analyzes in the results.

Otherwise, we enjoyed this article. We appreciate the opportunity to review chronobiological sciences.

We appreciate the immense help and excellent suggestions for improving our study
